# Novel Therapeutic Strategies in the Topical Treatment of Atopic Dermatitis

**DOI:** 10.3390/pharmaceutics14122767

**Published:** 2022-12-10

**Authors:** Lorenzo Maria Pinto, Andrea Chiricozzi, Laura Calabrese, Maria Mannino, Ketty Peris

**Affiliations:** 1UOC di Dermatologia, Dipartimento di Scienze Mediche e Chirurgiche, Fondazione Policlinico Universitario A. Gemelli—IRCCS, 00168 Rome, Italy; 2Dermatologia, Dipartimento di Medicina e Chirurgia Traslazionale, Università Cattolica del Sacro Cuore, 00168 Rome, Italy

**Keywords:** atopic dermatitis, targeted therapies, topical treatment, pipeline

## Abstract

Topical agents that are currently available for the treatment of atopic dermatitis may represent a valid approach in the management of mild or mild–moderate cases, whereas they are often supplemented with systemic therapies for handling more complex or unresponsive cases. The most used compounds include topical corticosteroids and calcineurin inhibitors, although their use might be burdened by side effects, poor response, and low patient compliance. Consequently, new innovative drugs with higher efficacy and safety both in the short and long term need to be integrated into clinical practice. A deeper understanding of the complex pathogenesis of the disease has led to identifying new therapeutic targets and to the development of innovative therapeutics. This narrative review aims to collect data on selected promising topical drugs that are in an advanced stage of development.

## 1. Introduction

Atopic dermatitis (AD) is an immune-mediated skin disease with a chronic relapsing course, characterized by eczematous lesions often associated with intense itching. It represents the most frequent inflammatory skin disease, with a worldwide prevalence ranging from 5–10% in children and 1–3% in adults [1,2]. Globally, an increased incidence of AD is emerging in Western industrialized countries as well as in low-income areas [3,4]. AD onset occurs before the first year of life, in approximately 50% of cases, and in 85% before the fifth year, while in 5% of individuals AD persists beyond adolescence [5]. Adult-onset AD accounts for 25% of all cases in adulthood [6].

The pathogenesis of the disease is complex, involving environmental and genetic factors, resulting in an altered immune response, impaired epidermal barrier function, along with cutaneous dysbiosis.

Clinical manifestations of AD are heterogeneous with various phenotypes that have been described [7]. Typical features include symmetrical, erythematous, and scaly papules or plaques, involving flexural areas, face, neck and distal extremities in adolescent and adults [8,9]. The pivotal symptom is chronic itching resulting in lichenification and excoriation, along with an increased risk of superimposed infections and a profound negative impact on the patient’s quality-of-life (QoL), also affecting sleep. Indeed, AD is associated with a high incidence of sleep disturbances, depression, and attention-deficit/hyperactivity disorder (ADHD), and frequently causes social isolation and work impairment [10,11]. Patients with AD present increased rates of comorbidities, such as cardiometabolic, neuropsychiatric, and malignant disorders [12]. Besides the negative impact on QoL, affecting both patients and their caregivers, AD is also associated with an unfavorable economic burden. Daily skin management is highly time-consuming, and direct and indirect economic efforts for therapeutic management may be particularly burdensome.

Multiple therapeutic options are currently available, ranging from topical approaches for mild disease (emollients, topical corticosteroids, and calcineurin inhibitors) to phototherapy and systemic agents for moderate-to-severe cases. Choosing between these therapies depends on the clinical features of AD lesions (erythema, papulation/edema, excoriation, and lichenification), the extent of skin involvement, the patient’s reported symptoms, preferences, and the physician’s experience [13,14].

Topical corticosteroids (TCS) still represent the major therapy for most patients with mild or localized AD, as they exert anti-inflammatory, antiproliferative, and vasoconstrictive effects by downregulating the genes encoding proinflammatory cytokines. These drugs, although effective, are burdened by potential side effects such as skin atrophy, telangiectasias, and striae, in case of prolonged use. Appropriate choice of potency and vehicle for TCS should be considered to optimize the patient’s compliance and minimize adverse events.

Topical calcineurin inhibitors (TCI) are steroid-sparing drugs, including tacrolimus 0.03% and 0.1% ointment, and pimecrolimus 1% cream. They act through the blockade of T cell activation by inhibiting IL-2 and IFN-γ synthesis. The most frequent adverse events include skin burning and itching that usually reduce over time. TCIs have a satisfactory efficacy and safety profile.

Because of new topical agents recently approved or currently in the pipeline for AD (Figure 1 and Figure 2), this review aimed to collect data on their safety and efficacy, conducting a search in the published literature through 20 September 2022 that included original articles (i.e., case reports, case series, and randomized controlled trials) using the PubMed database, and clinical trial data published on ClinicalTrials.gov. The terms used for the PubMed search were as follows: “Atopic dermatitis” AND “topical treatment”, OR “target therapy”, OR “topical JAK inhibitors”, OR “pipeline”, OR “crisaborole”, OR “Aryl hydrocarbon Receptor modulator”, OR “PDE4 inhibitors”, OR “skin microbiome”.

The authors selected articles that outlined pivotal and novel insights into AD, with special regard to topical treatments.

## 2. AD Pathogenesis

AD is characterized by a multifactorial pathophysiology, resulting from the complex interaction between genetic factors, environmental influences, immunological system dysregulation, and the impairment of skin barrier function.

The complex immunological activation includes cells belonging to type 2 inflammatory cells, such as T helper 2 cells (Th2), T cytotoxic 2 cells (Tc2), type 2 innate lymphoid cells (ILC2s), mast cells and eosinophils, and other immunologic pathways driven by Th22, Th17/IL-23, and Th1 cytokines [15], depending on various factors, such as disease stage and ethnicity [16]. Activated type 2 inflammatory cells promote the release of proinflammatory cytokines, mainly interleukin (IL)-4, IL-13, IL-31, mediating skin inflammation, pruritus, and antigen-specific IgE production by B cells and plasma cells.

Overall, most of the soluble mediators involved in the pathogenesis of AD exert their effects upon binding to specific transmembrane receptor and activation of the downstream intracellular Janus kinase (JAK)/signal transducer and activator of transcription (STAT) signaling pathway. Four isoforms of JAK, all with tyrosine kinase (TYK) activity, are currently known: JAK1, JAK2, JAK3, and TYK2 [17]. Following the interaction between the ligand and its specific receptor, activation of associated JAKs occurs, which phosphorylate the intracellular domain of the receptor, creating a docking site for STATs. STATs translocate into the nucleus and modulate the transcription of target genes involved in cell proliferation, differentiation, and activation [18]. AD is associated with increased signaling through all four JAKs [19], although the ultimate effect of the activation of this pathway depends on the isoform being activated. For instance, IL4 and IL13 mainly induce the activation of JAK1 and JAK3, and subsequently of STAT6.

Due to the crucial role of the activation of the JAK/STAT pathway in AD pathogenesis, its inhibition through JAK inhibitors (JAKi) represents a current target of recently approved drugs, and several others are in the pipeline, with encouraging results (Table 1 and Table 2) [20].

Interestingly, although AD has been traditionally considered a T cell-mediated disease, recent evidence has pointed to the role of B cells and humoral immunity in its pathogenesis [21]. Activated B cells are thought to contribute to AD pathophysiology through their differentiation into plasma cells secreting IgE antibodies as well as acting as antigen-presenting cells (APC), activating the T cell-mediated immune responses [22]. Both IL-4 and IL-13, the two key cytokines in AD pathogenesis, can directly act on B cells and induce a selective class-switch production of IgE [23]. High serum levels of IgE were indeed reported in AD patients and correlated with AD severity [24]. Furthermore, a newly described subset of B cells, exerting a negative regulation on immune responses through IL-10 production (Breg cells), have been shown to play a role in AD pathogenesis [25]. Notably, levels of IL-10-producing B cells in peripheral blood have been reported to be significantly lower in AD patients than in healthy individuals, suggesting that depletion of this cell population may contribute to AD inflammation and severity [26]. Finally, the role of B cells in AD pathogenesis has also been suggested by some reports describing an improvement of AD lesions in patients treated with rituximab, an anti-CD20 monoclonal antibody that depletes CD20+ B cells [27].

## 3. Recently Approved Topical Drugs for AD

### 3.1. Crisaborole

Crisaborole is a specific inhibitor of phosphodiesterase 4 (PDE4), an enzyme that plays a key role in regulating proinflammatory cytokine production. Indeed, PDE4, through degradation of cyclic adenosine monophosphate (cAMP), results in the inhibition of protein kinase type A (PKA) and activation of nuclear factor κB (NF-κB) and nuclear factor of activated T cells (NFAT), with a subsequent increase in specific cytokines such as (TNF)-alpha, IL-5, IL-10, and IL-2 [28,29]. This molecular process would explain the importance of PDE4 in the pathogenesis of AD, and therefore the therapeutic relevance of its inhibition.

Crisaborole 2% ointment was originally approved in 2016 by the Food and Drug Administration (FDA), for the treatment of moderate-to-severe AD in children ≥2 years old, and in March 2020 it obtained a therapeutic extension for patients aged >3 months [30,31].

The efficacy and safety of crisaborole were assessed in a large patient cohort enrolled in two randomized, double-blind, vehicle-controlled, phase III trials (AD-301; AD-302), including a total of 1522 patients ≥ 2 years of age with mild-to-moderate AD. The primary endpoint, meant as Investigator’s Static Global Assessment (IGSA) 0/1 at day 29, was achieved by a higher percentage of patients in the crisaborole arm vs. control (AD-301: 32.8% vs. 25.4%, *p* = 0.038; AD-302: 31.4% vs. 18.0%, *p* = 0.001).

Earlier reduction in itch emerged in patients treated with crisaborole compared with placebo (pooled data, 1.37 vs. 1.70 days, *p* = 0.001) [32]. This improvement in itch sensation was maintained throughout the duration of the study and confirmed by a subsequent analysis of the phase III trials performed by Yosipovitch et al. (56.6% vs. 39.5%; *p* < 0.001) [33,34]. Crisaborole also demonstrated a satisfactory safety profile, with 10.3% of enrolled patients developing adverse events, including AD (3.1%), application-site pain (2.3%), and application-site infection (1.2%).

A phase IV open-label study (CrisADe CARE 1) evaluated the efficacy and safety of crisaborole in patients aged 3–24 months with mild-to-moderate AD, and the primary endpoint (IGSA 0/1) was met by 30.2% of patients, with a mean percentage change in the Eczema Area and Severity Index (EASI) score of −57.5% [35].

### 3.2. Delgocitinib

The nonselective pan-JAKi, delgocitinib, in ointment formulation, was approved in Japan in 2020 for the treatment of AD [36]. The randomized, double-blind, phase III trial included Japanese patients ≥16 years with moderate-to severe AD. Participants were randomized in a 1:1 ratio to delgocitinib 0.5% ointment or vehicle twice daily for four weeks. The modified EASI (mEASI)-75 was achieved by 26.4% of delgocitinib-treated patients, compared with 5.8% in the placebo group (*p* < 0.01). The mEASI is a modified measure that excludes the assessment of untreated body areas in the study. Efficacy data were confirmed in a 52-week extension study [37,38]. Similarly to mEASI, IGA and itch numeric rating scale (NRS) scores were significantly reduced in the delgocitinib arm compared with the placebo.

Side effects were reported in 69% of patients, the most common adverse events were being nasopharyngitis (25.9%) and contact dermatitis (4.5%). Comparable efficacy data were obtained from a phase III trial enrolling patients aged 2–15 years. Patients applied delgocitinib 0.25% ointment for four weeks, with a favorable safety profile [39]. Delgocitinib cream (LEO242549) is currently being tested in a phase III trial for the treatment of chronic hand eczema (CHE), whereas the investigation for AD has been discontinued in both Europe and USA [40].

### 3.3. Ruxolitinib

Ruxolitinib 1.5% cream selectively inhibits JAK1 and JAK2, and was approved by the FDA in September 2021 for short-term and noncontinuous chronic treatment of mild-to-moderate AD in immunocompetent patients ≥12 years [41].

Two parallel, randomized, double-blind, phase III clinical trials (Topical Ruxolitinib Evaluation in Atopic Dermatitis: TRuE-AD1 and TRuE-AD2), investigated the efficacy and safety of topical ruxolitinib 0.75% and 1.5% cream for 8 weeks. A total of 1249 mild-to-moderate AD patients aged ≥12 years with a baseline IGA 2/3 and body surface area (BSA) of 3–20%, were enrolled in the study. In both trials, a greater number of patients achieved an IGA 0/1 and an improvement of ≥ 2 grades compared with baseline using ruxolitinib at 0.75% (50% TRuE-AD1, 39.0% TRuE-AD2) or 1.5% (53.8%, 51.3%) compared with the vehicle group (15.1%, 7.6%) [41]. A rapid and prolonged amelioration of itch was detected using ruxolitinib 1.5% or 0.75% cream in both the TRuE-AD1 and TRuE-AD2 trials. A rapid efficacy in reducing pruritus was also supported by a recent subanalysis derived from TRuE-AD trial program data [42].

Ruxolitinib was generally well tolerated in both treatment arms, with nasopharyngitis being the most common adverse event reported, whose incidence was significantly higher (3% and 2.6%) in ruxolitinib arms compared with placebo (0.8%) [43]. Treatment-related adverse events included folliculitis at the application site and acne.

The efficacy of ruxolitinib was also assessed among adolescents, demonstrating an EASI ≥ 90% improvement (EASI-90) in 41.5% of patients applying 0.75% cream, 39.1% for 1.5% ruxolitinib cream, and 7% for placebo, with a safety profile in line with previous trial outcomes [44].

A recent pooled analysis proved similar efficacy in terms of IGA 0/1 achievement in the Black or African American population, and as a global improvement across all body areas [45,46].

Several studies are currently ongoing, specifically a phase III, double-blind, randomized, vehicle-controlled study followed by a long-term safety extension period in children with AD (age ≥2 years to <12 years) (TRuE-AD3). A phase III, open-label, one-year safety study of ruxolitinib cream in adolescents with AD (age ≥12 < 18 years) is not yet recruiting. Localized facial and/or neck AD efficacy and safety will be investigated in a phase II trial, actively recruiting.

## 4. New Emerging Topical Treatments for AD in the Pipeline

### 4.1. Aryl Hydrocarbon Receptor Modulating Agents

Aryl hydrocarbon Receptor (AhR) is a ligand-dependent receptor that acts as a transcription factor, and it is expressed on the surface of several immune cells and epithelial cells, including the skin, intestine, and lungs. Several molecules, both endogenous and exogenous, can act as AhR ligands, leading to its activation. AhR is involved in the transcriptional regulation of multiple genes implicated in the immune response, cellular apoptosis, proper functioning of anti-inflammatory regulatory T cells (Tregs), pro-inflammatory cytokine suppression, and keratinocytes terminal differentiation [47]. A deeper understanding of the molecular processes mediated by AhR has made it possible to develop a targeted therapy for this pathway.

Tapinarof 1% cream (GSK2894512 cream, WBI-1001, or Benvitimod^TM^) is an AhR modulating agent for topical use, under development for the treatment of AD and psoriasis [48,49]. A phase IIb, double-blind, vehicle-controlled randomized study included adolescents and adults (aged 12–65 years) with AD to receive tapinarof cream 0.5%, 1%, or vehicle, once or twice daily for 12 weeks, with a four-week treatment-free follow-up [50,51]. The percentage of patients achieving IGA response was higher for all tapinarof groups than for the vehicle at each visit by week 2 and maintained throughout the follow-up period. Tapinarof 1% cream-treated patients demonstrated higher response rates than the 0.5% group. At the last observation, IGA response rates were 53% (1% twice daily; *p* = 0.008) and 37% (0.5% twice daily; *p* = 0.240) vs. 24% (vehicle twice daily). At week 12, the mean change in EASI score was significantly higher in all tapinarof groups than in the vehicle group, and this was evident as early as week 1: 73% (1% twice daily; *p* < 0.001) and 66% (0.5% twice daily; *p* < 0.004) vs. 38% (placebo twice daily). Tapinarof was well tolerated, with most adverse events evaluated as mild or moderate, mainly represented by folliculitis. Phase III clinical trial programs (ADORING 1 and ADORING 2) are two 8-week, parallel studies currently ongoing in children aged ≥2 years and adults with moderate-to-severe AD, and will be followed by an open-label, 48-week extension study (ADORING 3) [52].

### 4.2. Phosphodiesterase 4 Inhibitors

The intracellular enzyme PDE4 plays a major role in AD pathogenesis and its inhibition may represent a valuable therapeutic strategy. In addition to **crisaborole**, a topical anti-PDE4 agent, other topical PDE4 inhibitors are in clinical development. Topical application avoids the typical gastrointestinal adverse events related to systemic administration.

**Roflumilast** cream (**ARQ-151**) is a highly potent PDE-4 inhibitor. Its efficacy and safety are under investigation for both plaque psoriasis (phase III study) and mild-to-moderate AD [53].

A four-week, phase II proof-of-concept study involved 136 AD patients ≥ 12 years, randomized to roflumilast 0.15%, 0.05%, or placebo. The primary endpoint, represented by the absolute change from baseline in EASI score at week 4, was not achieved.

Two parallel, double-blind, vehicle-controlled phase III trials (Integument-1 and Integument-2), undertaken in February 2021, evaluated patients ≥6 years, with AD involving >3% BSA, treated once daily with roflumilast 0.15% for 4 weeks [54,55].

A phase III trial (Integument-PED) is currently investigating the application of roflumilast 0.05%, in patients between 2 and 5 years of age, with a total of 650 patients enrolled, posing as primary outcome IGA success, defined as vIGA-AD score 0/1 and 2-grade improvement from baseline at week 4 [56]. Results from phase III trials are not yet available.

**Difamilast** is a topical selective PDE4 inhibitor tested in two phase III trials that have been recently completed in Japan for both adult and pediatric populations [57]. In the first one, patients aged 15–70 years with an IGA of 2/3, treated with difamilast ointment 1% or placebo twice daily for four weeks, were included. The percentage of patients on difamilast therapy who achieved an IGA 0/1 score at week 4 was higher than the placebo group (38.46% vs. 12.64%, respectively, *p* < 0.0001). At week 4, EASI-75 was obtained in 42.86% of difamilast-treated patients vs. 13.19% of placebo (*p* < 0.0001), while EASI-90 was observed in 24.73% vs. 5.49%, respectively (*p* < 0.0001) [58].

Treatment-emergent adverse events were observed in 32 (28%) of difamilast-treated patients vs. 51 (28%) of the placebo arm, and the most frequently encountered were AD (1% difamilast 3.8%; vehicle 12.1%) and nasopharyngitis (1% difamilast 4.9%; vehicle 3.8%). The main treatment-emergent adverse event that led to treatment discontinuation was AD worsening (1% difamilast: 3.8%; placebo: 9.3%).

The efficacy and safety of difamilast 0.3% and 1% on a pediatric population were evaluated in a randomized, double-blind, vehicle-controlled phase III study. Patients between 2 and 14 years of age with IGA 2/3 were enrolled and randomized in three arms: difamilast 0.3% (n = 83), difamilast 1% (n = 85), or vehicle (n = 83) applied twice daily, for 4 weeks. The primary endpoint was the percentage of patients with an IGA of 0/1 and an improvement by at least two grades at week 4. Success rates in IGA scores at week 4 were higher in the active arms (44.6% and 47.1% vs. 18.1%). Therapeutic success, meant as EASI 75, at week 4 was detected in 43.4% of the difamilast 0.3% group and 57.7% of the difamilast 1% group vs. 18.1% of the vehicle group. The favorable safety profile resulting from this study is comparable to that previously established [59,60].

A phase III, multicenter, open-label, uncontrolled study evaluating the efficacy and safety of difamilast ointment in infants younger than two years of age is currently ongoing. At the end of 2021, difamilast ointment received approval in Japan for the treatment of AD patients ≥2 years [61].

An additional agent named **LEO 29102** selectively blocks the PDE4D isoform [62]. A phase II proof-of-concept and dose-finding study enrolled 183 patients, splitting them into differently-dosed arms and comparing LEO 29102 with pimecrolimus and placebo.

However, due to the lack of statistical significance in terms of both pruritus and EASI reduction compared to baseline, the drug development was discontinued [63].

The evaluation of the efficacy and safety of **lotamilast** (RVT-501/E6005) was the aim of two Japanese clinical trials in adult and pediatric populations with mild–moderate AD [64,65]. Seventy-eight adults were randomized to lotamilast 0.2% or vehicle for 4 weeks followed by an extension period of 8 weeks. The evaluation of clinical efficacy showed an improvement, although not statistically significant, in the treated group compared to placebo, in terms of EASI score, Severity Scoring Atopic Dermatitis (SCORAD)-objective, SCORAD-C (visual analog scales for pruritus and sleep loss), and itch NRS.

Regarding the extension study, the group receiving topical E6005 for 12 weeks showed significant score reductions from baseline for EASI (*p* = 0.030), SCORAD-objective (*p* < 0.001), and SCORAD-C (*p* = 0.038).

In pediatric patients, a greater decrease in severity score was obtained in the lotamilast-treated group compared to the vehicle (−45.94% vs. −32.26%), although not statistically significant.

A multicenter, randomized, vehicle-controlled, double-blind, phase II trial involving adult and adolescent subjects with mild-to-moderate AD was designed to evaluate safety, efficacy, and pharmacokinetics of 0.2% and 0.5% concentrations of **RVT-501**, but results are not yet available [66].

A phase II clinical study is currently underway to study the efficacy and safety of **Hemay808** at various concentrations (1%, 3%, 7%) for the treatment of adults with mild-to-moderate AD.

A phase IIa, randomized, double-blind, vehicle-controlled, parallel group study recently published the results of a group analysis concerning the efficacy and safety of **PF-07038124** ointment in adult patients, with mild-to-moderate AD or plaque psoriasis. The study enrolled 104 participants, 70 for the AD group and 34 for plaque psoriasis, randomized to PF-07038124 0.01% or placebo. Active-arm AD participants achieved a mean percent change from baseline to week 6 in EASI score of −74.9% vs. −35.5% in the placebo group (*p* = 0.0004). A total of 44.4% of AD patients in the treatment group achieved an IGA 0/1 or two-point reduction at week 6 compared with baseline (vs. placebo, 8.8%).

### 4.3. JAK Inhibitors

In addition to the topical JAKi already approved, others are currently under investigation.

**Cerdulatinib** (DMVT-502) simultaneously blocks all JAKs and spleen tyrosine kinase (SYK) [67]. A phase Ib, single-center, double-blind study evaluated the safety and efficacy of cerdulatinib 0.37% gel and its molecular effects on lesional skin in adults with mild-to-moderate AD. Mean EASI scores decreased significantly from baseline to day 14, with a mean change of −2.6. Adverse events were generally mild, occurring in 8/10 cerdulatinib-treated patients, and headache was the most common adverse event reported. To investigate the effects of the drug on epidermal thickness, cell infiltration, and gene expression, lesional skin biopsies were obtained at baseline and after 14 days of treatment. The results showed a reduction in both epidermal thickness and expression of key inflammatory markers, and a decrease in inflammatory dendritic cell infiltration [68].

**ATI-1777** (JAK1/3 inhibitor) completed a phase IIa trial with positive results, as shown by preliminary released data [69]. During a 4-week treatment period, 50 adults with moderate-to-severe AD were randomized in a 1:1 ratio of ATI-1777 2.0% topical solution or placebo applied twice daily. A 74.4%-reduction in the mEASI score from baseline to week 4 emerged in subjects applying ATI-1777 (vs. 41.4% of placebo; *p* <0.0001), with minimal systemic exposure. Improvements in itch and BSA reduction were also observed. Subsequently, a phase IIb, multicenter, randomized, double-blind, vehicle-controlled study was performed to determine the safety, tolerability, pharmacokinetics, and efficacy of ATI-1777 in patients aged 12–65 years old with moderate-to-severe AD.

**ATI-502** (JAK1/3 inhibitor) was evaluated in an 8-week phase IIa safety and tolerability study in adults with moderate-to-severe AD [70]. No other trials are ongoing.

**SHR0302** is a highly selective JAK1 inhibitor, investigated in both oral and topical formulations [71]. A phase II/III, randomized, double-blind, vehicle-controlled, seamless study is ongoing to evaluate the efficacy and safety of topical SHR0302 at different doses in adult and adolescent patients with mild-to-moderate AD.

**Brepocitinib** (JAK1/TYK2), completed a phase IIb vehicle-controlled trial with a primary endpoint meant as the change from baseline in EASI score at week 6, with different drug concentrations. Brepocitinib 1% QD and 1% BID treatment arms achieved a statistically significant difference compared to the placebo at week 6. The safety profile appeared favorable [72].

Other topical JAKi, namely, jaktinib (pan-JAKi), and CEE321 (pan-JAKi), are currently under clinical investigation and the results are not yet available.

**Tofacitinib** is a JAK1/3 inhibitor [73] whose topical efficacy was investigated in a phase IIa trial with encouraging results (82% EASI reduction vs. 30%; prompt improvement of pruritus) and a good safety profile [74]. However, the clinical trial program was discontinued.

### 4.4. Skin Microbiome Modulating Agents

The role of the cutaneous microbiome is known as a pathogenetic factor contributing to AD, and therapeutic interventions on the microbiome would likely lead to clinical benefits for patients with AD. Current research is focused on the development of topical drugs capable of restoring its composition, via bacterial replacement and microbiome transplantation (Table 2) [75,76].

**FB-401** is a compound under development that combines three chains of *Roseomonas mucosa*, a Gram-negative bacterium that appears predominant in AD patients’ skin. Its activity seems to be related to the activation of tissue repair systems and anti-inflammatory activity via Toll-like receptor 5 (TLR5) and tumor necrosis factor receptor (TNFR) [75].

A randomized, double-blind, placebo-controlled, 16-week phase II trial was designed to evaluate the efficacy and safety of FB-401 in children and adults with AD. Despite the encouraging results obtained (60% of adults showed a 50% reduction in the SCORAD values, 90% of pediatric patients achieved EASI50) [77], the phase II study was discontinued due to the lack of a statistically significant difference in achieving the primary outcome (EASI 50, 58% in FB-401 patients vs. 60% in placebo; *p* = 0.7567).

AD-like skin inflammation was triggered in mice models by certain strains of *Staphylococcus aureus* isolated from patients with severe AD. Topical application of a lyophilized human-derived strain of ***Staphylococcus hominis A9 (ShA9)*** should inhibit *S. aureus* overgrowth and toxin production, allowing nonpathogen, commensal microbiome to proliferate [78]. To assess the safety, mechanism of action, and potential therapeutic efficacy of ShA9, a randomized, double-blind, vehicle-controlled phase I study was designed. The study drug was applied for one week on the forearm skin of 54 adults with *S. aureus* cultured-positive AD. No serious adverse events were reported in the ShA9-treated arm, compared with the placebo. Overall, adverse events were mild and fewer in the active arm (*p* = 0.044). No significant difference between the two groups was observed in terms of clinical efficacy assessed through the EASI score and SCORAD. Another phase 1 open-label study is currently recruiting participants with moderate-to-severe AD to assess the survival of skin-transplanted ShA9 on lesional and non-lesional AD skin [79]. All patients will receive the study compound and the vehicle on their arm.

**Nitrosomonas eutropha** (B244), an ammonia-oxidizing bacteria (AOB), produces nitric oxide, which has anti-inflammatory and vasodilator capabilities, and nitrite, with anti-infective activity [80]. It also downregulates the production of IL-4, IL-5, IL-13, and IL-31 [72]. B244 is frequently deficient in the skin, so the aim of this novel potential compound is to re-establish the normal composition of the cutaneous microbiome. Phase II trials have been completed both on atopic dermatitis and pruritus, with promising results and no safety issues.

**Niclosamide (ATx201)** is an investigational drug delivered topically and designed to reduce colonization by *S. aureus*, known to be linked to disease severity and progression. A randomized, double-blind, placebo-controlled phase II trial investigated ATx201 ointment 2% applied twice daily in patients with mild-to-severe AD [81]. The study drug reduced *S. aureus* colonization (94.4% success rate vs. 38.9%, *p* = 0.0016) and significantly increased Shannon diversity in the skin microbiome at day 7.

**Omiganan pentachloride** is a synthetic indolicidin analog peptide that inhibits *S. aureus* overgrowth and exerts anti-inflammatory effects [82], tested in a phase II study with the aim of assessing its pharmacodynamics, tolerability, and efficacy [83]. Although the compound was effective in reducing the proliferation of *S. aureus*, the 28-day treatment was not associated with an amelioration of clinical manifestations in patients with mild-to-moderate AD.

### 4.5. Other Novel Therapies in the Pipeline

Currently, many other topical compounds are in the pipeline for AD with distinct mechanisms of action, and some promising molecules in the later stage of development are discussed below. Among these, it is worth mentioning the liver X receptors (LXRs), ligand-activated nuclear transcription factors, with a central role in maintaining skin barrier homeostasis and regulating the immune response.

The LXR-β agonist, **VTP-38543** cream, was investigated in a double-blind, placebo-controlled phase II study including 104 patients with mild-to-moderate AD, at different concentrations (0.05%, 0.15%, and 1.0%) applied twice daily for 28 days. The aim of this study was to evaluate its safety, tolerability, and clinical efficacy, assessed by SCORAD, IGA score, and EASI. Thirty-three patients underwent skin biopsy prior to and after treatment to examine the cutaneous expression of biomarkers [84]. The drug was safe and well tolerated. Skin biopsies demonstrated an improvement in the skin barrier, with an increased expression of mRNA encoding for loricrin and profilaggrin, reduced epidermal thickness, and nonsignificant reduction in Th2/Th17 markers.

Another LXR-β agonist, **ALX 101,** completed its phase IIb study, but the results are not yet available [85].

**BEN-2293,** a pan-tropomyosin receptor kinase (pan-TRK) antagonist, is currently under investigation in a double-blind, placebo-controlled phase I/II proof-of-concept study to assess the safety, tolerability, pharmacokinetics, and preliminary efficacy of the topical ointment (0.25% or 1% concentration) [86].

**AMTX-100** is a cell-penetrating nuclear transport modifier (NTM), effective in modulating the transport of transcription factors (NF-κB, NFAT, AP-1, and STAT1) that regulate the expression of genes involved in the production of inflammatory cytokines and chemokines (TNFα, IL-1β, IL-6, IL-17, etc.) [87]. A two-part, phase I/IIb, multicenter, double-blind, randomized, vehicle-controlled trial has been designed to evaluate the safety and efficacy of topically applied AMTX-100 in adults with mild-to-moderate disease.

Another frontier that is currently being explored is focused on the covalent inhibition of Bruton tyrosine kinase (BTK), a key intracellular signaling factor involved in immune responses [88]. The presently approved drugs within this class have oncological indications and lead to the irreversible inhibition of BKT. Topical **PRN473** showed sustained and potent reversible inhibition of BTK along with low systemic exposure in preclinical studies. Its mechanisms of action include downregulation of signaling for BCR, Fc receptors, and specifically inhibited IgE (FcεR)-mediated mast cell and basophil activation, IgG (FcγR)-mediated monocyte activation, and reduced neutrophil migration [89]. PRN473 is being explored for the treatment of AD in a phase IIa trial.

**AKP-11** is a topical highly selective Spingosine-1-Phosphate receptor subtype-1 (S1PR1) modulator. Sphingosine 1-phosphate (S1P) is a ceramide-derived lipid via ceramidase and sphingosine kinase activity. S1P binds to high-affinity G-protein-coupled receptors, S1RPs, modulating cell proliferation and differentiation, proinflammatory cytokine production, proper skin barrier function, migration, and degranulation of endothelial, epithelial, nervous, and immune cells [90]. AKP-11, via its modulation of S1PR1, proved to be safe and well-tolerated in two phase-1 studies in subjects with psoriasis and AD. Its efficacy and safety will be evaluated in phase II trials in patients with AD and prurigo nodularis.

## 5. Discussion

Despite TCS and TCI remain the mainstay of the topical approach for AD in a real-world setting, novel topical agents are being developed in order to fulfill important unmet needs represented by (i) the limited number of compounds with immunomodulant mechanisms that could represent a valid alternative strategy to corticosteroids; (ii) efficacy of therapeutic options that could be comparable with corticosteroids; and (iii) ease of use through innovative and different formulations that could enhance patients’ adherence and optimize drug delivery (i.e., nanomolecules). A deeper understanding of AD pathogenesis has led to the development of topical compounds targeting different molecules, mainly involved in the regulation of the abnormal immune and inflammatory response, homeostasis of the skin barrier, and modulation of the cutaneous microbiome. The most promising ones include AhR, PDE4, JAK/STAT inhibitors, and skin microbiome modulating factors.

Tapinarof 1% cream, an AhR modulator that acts as a transcription factor, has successfully completed phase IIb trials, proving to be highly effective and well tolerated in patients affected by moderate-to-severe AD. It is currently under development in a phase III program, which also involves pediatric subjects aged ≥2 years.

Several PDE4 inhibitors are in the pipeline following FDA approval for crisaborole 2% ointment, for AD patients aged ≥ 3 months. Difamilast recently received manufacturing and marketing approval in Japan for the treatment of both adults and children with mild-to-moderate disease, after positive results achieved in phase III trials.

JAK inhibitors, available in both topical and oral formulations, are proving to be highly effective agents in the management of AD. In addition to the approved ruxolitinib and delgocitinib, several other compounds are in late-stage development phases.

Multiple strategies for restoring the skin microbiome have been proposed and this therapeutic approach, one of the most interesting future directions in the AD pipeline. Despite the considerable interest in AD, the pathogenic relevance of skin dysbiosis in AD remains unclear, and, therefore, the pharmacological potentiality of these drugs is not fully established.

Although some limitations of the therapeutic approach in terms of long-term use might be foreseen (low patients’ adherence due to a daily time-consuming approach), the future topical agents may offer promising perspectives regardless of whether advantages over the currently available topical compounds, such as higher efficacy, long-lasting effect after treatment withdrawal, and a marked reduction in side effects related to the long-term, will be demonstrated.

## 6. Conclusions

The topical approach is a cornerstone in the therapeutic armamentarium for AD but consists of a limited number of agents. A better understanding of AD pathogenesis is leading to the identification of therapeutic targets and, thus, the development of new drugs that could expand the opportunity to manage AD manifestations with multiple lines of topical compounds. Furthermore, a greater variety of topical compounds would allow tailoring the topical therapeutic approach, enabling the implementation of precision medicine in the daily management of AD patients.

## Figures and Tables

**Figure 1 pharmaceutics-14-02767-f001:**
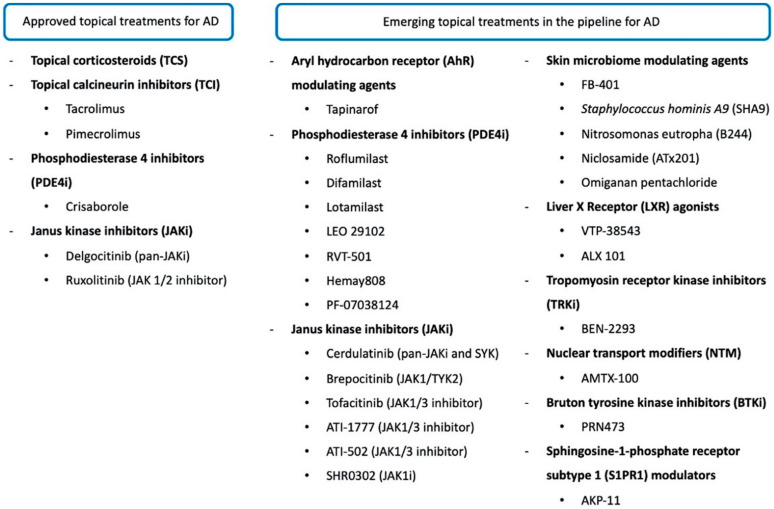
Currently approved and emerging topical treatments for atopic dermatitis. Abbreviations: AD, atopic dermatitis; SYK, spleen tyrosine kinase; TYK2, tyrosine kinase 2.

**Figure 2 pharmaceutics-14-02767-f002:**
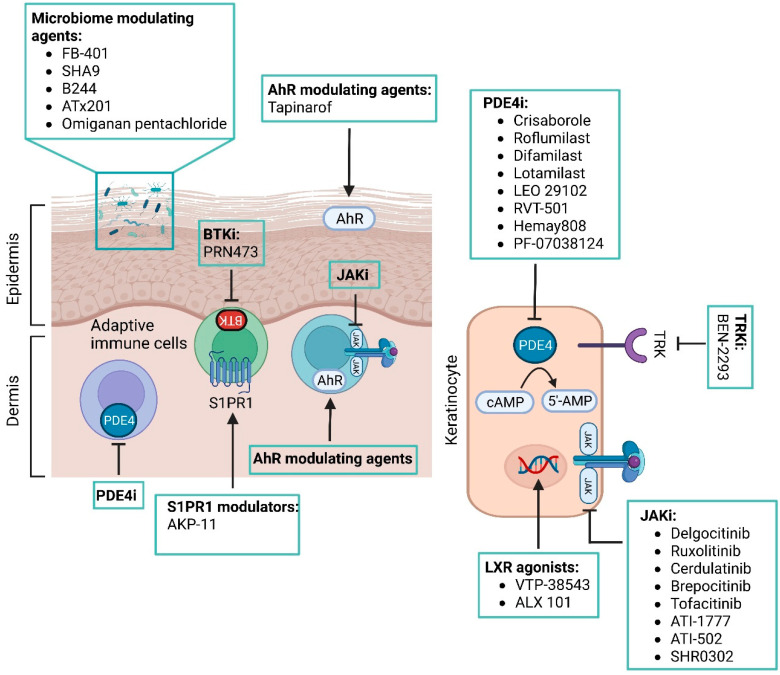
Outline of topical therapeutic strategies and relevant cellular targets in atopic dermatitis. Abbreviations: AhR, aryl hydrocarbon receptor; BTK, Bruton tyrosine kinase; BTKi, Bruton tyrosine kinase inhibitors; cAMP, cyclic adenosine monophosphate; JAK, Janus kinase; JAKi, Janus kinase inhibitors; LXR, liver X receptor; PDE4, phosphodiesterase 4; PDE4i, phosphodiesterase 4 inhibitors; S1PR1, sphingosine-1-phosphate receptor subtype 1; TRK, tropomyosin receptor kinase; TRKi, tropomyosin receptor kinase inhibitors. Created with BioRender.com.

**Table 1 pharmaceutics-14-02767-t001:** Topical treatments in the pipeline for atopic dermatitis in the late stage of development.

	Agent (Acronym)	Inhibition Target/Activity	Clinical Development Phase	Clinical Trial Identifier
Aryl Hydrocarbon Receptor Modulating Agents	Tapinarof/benvitimod	AhR agonist	III	NCT05142774, NCT05014568, NCT05032859
Phosphodiesterase 4 Inhibitors	Roflumilast (ARQ-151)	PDE4	III	NCT04804605, NCT04773600, NCT04845620, NCT04773587
	Difamilast (OPA-15406/MM36)	PDE4	III	NCT05372653, NCT03908970, NCT03911401
	Lotamilast (RVT-501/E6005)	PDE4	III	NCT03394677, NCT02950922
	PF-07038124	PDE4	III	NCT05375955
Topical JAK inhibitors	Delgocitinib	Pan-JAK	Approved in Japan; IIb in EU	NCT03725722
	Ruxolitinib	JAK1, JAK2	III	NCT04921969, NCT05456529, NCT03745638, NCT03745651

Abbreviations: AhR, aryl-hydrocarbon receptor; PDE4, phosphodiesterase 4; JAK, Janus kinase.

**Table 2 pharmaceutics-14-02767-t002:** Other novel topical therapies currently in phase I/II of development for atopic dermatitis.

	Agent (Acronym)	Inhibition Target/Activity	Study Phase	Clinical Trial Identifier	AD Severity	Study Duration	Age (Years)	Primary Endpoint	Status
Skin microbiome modulating agents	FB-401	Bacterial replacement	IIb	NCT04504279	Mild to moderate	16 weeks	≥2	EASI50	Completed
	*Sh*A9	Targeted microbiome transplant	I	NCT05177328	Moderate to severe	24 days	18–80	Duration of ShA9 survival on the lesional ventral arm skin	Recruting
	*Nitrosomonas eutropha* (B244)	Nitric oxide donor	IIb	NCT04490109	Mild to moderate	4 weeks	18–65	Mean change in WI-NRS	Completed
	Niclosamide (ATx201)	Decolonization of *S. aureus*	II	NCT04339985	Mild to moderate	2 weeks	12–60	Mean change in EASI score	Completed
	Omiganan pentachloride	Antimicrobial cationic peptide	II	NCT03091426	Mild to moderate	7 weeks	18–65	Clinical evaluation [oSCORAD]	Completed
Others	VTP-38543	LXR-β agonist	I/II	NCT02655679	Mild to moderate	28 days	18–65	Number of AEs	Completed
	ALX 101	LXR-β agonist	II	NCT03859986	Moderate	56 days	≥12	Mean change in EASI score at week 8	Unknown
	BEN-2293	Pan-TRK	I/II	NCT04737304	Mild to moderate	NA	18–65	Safety and tolerability	Recruiting
	HY209	GPCR19	II	NCT04530643	Mild to moderate	28 days	18–65	Safety and tolerability	Recruiting
	AMTX-100	Nuclear transport modifier	I/IIb	NCT04313400	Mild to moderate	NA	≥18	Maximum tolerable dose	Active not recruiting
	PRN473	BTK inhibitor	IIa	NCT04992546	Mild to moderate	42 days	18–70	Number of AEs	Recruiting
	AKP-11	S1PR1	II	NA					
Phosphodiesterase 4 Inhibitors	Hemay-808	PDE4	II	NCT04352595	Mild to moderate	29 days	18–65	Change in the EASI score relative to the baseline	NA
	LEO 29102	PDE4	II	NCT01037881	Mild to moderate	28 days	18–65	Mean change in EASI score at week 8	Completed
Topical JAK inhibitors	Tofacitinib	JAK1, JAK3	II (discontinued)	NCT02001181	Mild to moderate	28 days	18–60	Change in the EASI score relative to the baseline	Completed
	ATI-1777	JAK1, JAK3	IIb	NCT05432596	Moderate to severe	28 days	12–65	Percentage change from baseline in EASI score at Week 4	Recruiting
	ATI-502	JAK1, JAK3	II (discontinued)	NCT03585296	Moderate to severe	56 days	>18	Number of AEs	Completed
	Brepocitinib (PF-06700841)	JAK1/TYK2	IIb	NCT03903822	Moderate to severe	42 days	12–75	Percentage change from baseline in EASI score at Week 8	Completed
	Jaktinib	Pan-JAK	I/II	NCT04435392	Mild to moderate	NA	18–65	Proportion of participants achieving PGA response of 0/1	Recruting
	CEE321	Pan-JAK	I	NCT04612062	Mild to moderate	NA	18–65	Number of AEs	Completed

Abbreviations: AEs, adverse events; EASI, eczema area severity index; PGAShA9, Staphylococcus hominis A9; LXR, liver X receptors; TRK, tropomyosin receptor kinase; NA, not applicable; GPCR19, G-protein-coupled receptor 19; BTK, Bruton tyrosine kinase. S1PR1, Spingosine-1-Phosphate receptor subtype-1.

## Data Availability

Not applicable.

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
