# Peer review of "Novel Therapeutic Strategies in the Topical Treatment of Atopic Dermatitis"

_pharmaceutics, 2022, doi:10.3390/pharmaceutics14122767_

Round 1

Reviewer 1 Report

The manuscript by Pinto et al. ‘Novel therapeutic strategies in the topical treatment of atopic dermatitis’ seems interesting and informative. However, the following points should be considered before publication.

Figures, or diagrams are lacking. Make review more diagrammatic. If possible, summarize current delivery strategies in diagrammatically. Also, illustrate the schematic diagram that might highlights the objective of this review.

Discuss/Include the limitations/challenges of existing therapeutic or delivery strategies for topical treatment of atopic dermatitis.

Discuss/Include future perspective and future research direction.

Author Response

We thank Reviewer 1 for the positive comments to our review article. Please find below the point-by-point responses to your comments.

Figures, or diagrams are lacking. Make review more diagrammatic. If possible, summarize current delivery strategies in diagrammatically. Also, illustrate the schematic diagram that might highlights the objective of this review.

Answer: We implemented the manuscript with 2 figures: Figure 1 resumes the approved and emerging topical treatments in AD while Figure 2 is a cartoon the nicely shows the inhibitory activity of several topical compounds within the AD pathogenic setting.

Discuss/Include the limitations/challenges of existing therapeutic or delivery strategies for topical treatment of atopic dermatitis.

Answer: We included the main limitations and challenges of current topical options on page 8 of 23, lines 42-49 (tracked version), discussing what are the relevant unmet needs that are not fulfilled by the current topical options

Discuss/Include future perspective and future research direction.

Answer: We included a brief general comments on the future perspectives that could be foreseen considering the pipeline of topical agents described in this review (page 9 of 23, lines 10-16 of the tracked version)

Reviewer 2 Report

This is a nice review of the novel therapeutic strategies in the topical treatment of atopic dermatitis. I think this paper will be very helpful to the reader. 

However, I think the authors should add more description in the pathogenesis section. Specifically, B cell regulation is a very interesting issue in the pathogenesis of AD. So, I strongly recommend that the authors would add more things regarding B cell regulation.

Author Response

We thank Reviewer 2 for the positive comment and words of appreciation. We agreed B cell compartment is relevant in AD pathogenesis and we included a description of the pathogenic role of  B cells that is not only related to IgE production (page 3 of 23, lines 16-27 of the tracked version). All changes are highlighted through the text

Round 2

Reviewer 1 Report

Manuscript has improved than before. Authors have corrected partially. However, following points need to be considered:

1) I found the figure legends, however, could not find figures. Please put the figures with legends in the manuscript where it should be appeared.

2) Put only the corrected manuscript file. The manuscript file contains two files and makes bulky files. Thereby, creates difficulty in reviewing the manuscript.

3) Font size of the manuscript is too small to read. Remove tracking mode and just keep the edited part by highlighting with the color.

4) Please organized the manuscript well.

Author Response

We re-submitted the amended file that also includes figures, as suggested.

Round 3

Reviewer 1 Report

Authors have corrected the manuscript as per my suggestions and looks better. Thus, I recommend for publication.